# Deep Learning Applied to Intracranial Hemorrhage Detection

**DOI:** 10.3390/jimaging9020037

**Published:** 2023-02-07

**Authors:** Luis Cortés-Ferre, Miguel Angel Gutiérrez-Naranjo, Juan José Egea-Guerrero, Soledad Pérez-Sánchez, Marcin Balcerzyk

**Affiliations:** 1Department of Computer Sciences and Artificial Intelligence, University of Seville, Avda. Reina Mercedes s/n, 41012 Sevilla, Spain; 2Hospital Universitario Virgen del Rocio, Avda. Manuel Siurot, 41013 Sevilla, Spain; 3Instituto de Biomedicina de Sevilla (Universidad de Sevilla—CSIC—Junta de Andalucía), 41013 Sevilla, Spain; 4Stroke Unit, Neurology Department, Hospital Universitario Virgen Macarena, 41009 Sevilla, Spain; 5Neurovascular Research Laboratory, Instituto de Biomedicina de Sevilla-IBiS, 41013 Seville, Spain; 6Department of Medical Physiology and Biophysics, University of Seville, 41009 Sevilla, Spain; 7Centro Nacional Aceleradores (Universidad de Sevilla—CSIC—Junta de Andalucía), 41092 Sevilla, Spain

**Keywords:** image detection, intracranial hemorrhage, deep learning, decision support system

## Abstract

Intracranial hemorrhage is a serious medical problem that requires rapid and often intensive medical care. Identifying the location and type of any hemorrhage present is a critical step in the treatment of the patient. Detection of, and diagnosis of, a hemorrhage that requires an urgent procedure is a difficult and time-consuming process for human experts. In this paper, we propose methods based on EfficientDet’s deep-learning technology that can be applied to the diagnosis of hemorrhages at a patient level and which could, thus, become a decision-support system. Our proposal is two-fold. On the one hand, the proposed technique classifies slices of computed tomography scans for the presence of hemorrhage or its lack of, and evaluates whether the patient is positive in terms of hemorrhage, and achieving, in this regard, 92.7% accuracy and 0.978 ROC AUC. On the other hand, our methodology provides visual explanations of the chosen classification using the Grad-CAM methodology.

## 1. Introduction

Spontaneous intracranial hemorrhage (ICH) occurs when a diseased blood vessel within the brain bursts, allowing blood to leak inside the brain. Several factors, like disruption of the blood-brain barrier and leakage of fluids and proteins, inflammatory cascade or sudden increase in intracranial pressure after bleeding, develop into a brain injury in the areas surrounding the blood spill.

Nearly 66% of all deaths caused by neurological diseases worldwide are related to hemorrhagic stroke [1]. According to [2], the overall incidence of spontaneous ICH worldwide is 24.6 per 100,000 people each year, and approximately half of ICH related mortality occurs within the first 24 h [3].

The severity and outcome of an ICH depend on its cause, location in the brain, size of the bleed, the amount of time that passes between the bleed and treatment, clinical presentation, etc. ICH usually occurs in particular parts of the brain, including the basal ganglia, cerebellum, brain stem, or cortex.

Damage can be severe and result in physical, mental, and task-based disability. Various types of ICH strike people of all ages. Although ICH is usually associated with older adults, it can also occur in the younger population. If not treated correctly and immediately, an ICH can cause disability or death. As a heterogeneous disease, image characteristics help to determine the cause, the prognosis and the treatment.

To understand brain bleeds, it is important to have a basic understanding of the different types, based on the location where the bleed occurs (for a detailed introduction to ICH, see, e.g., [4] or [5]). After a stroke, the cause (blood or blood clot) must be determined to initiate appropriate treatment. Short-term medical treatment can help reduce brain damage and improve the likelihood of recovery. Survival and recovery from intracranial haemorrhage are related to the site, mass effect, and intracranial pressure of the underlying hematoma, and by any subsequent cerebral edema from perihematomal neurotoxicity or inflammation and complications from long-term neurological dysfunction.

Depending on the type, location, and extent of the brain bleed, many patients do not survive the initial bleeding event. Mortality rate is as high as 50% in the first 30 days [6]. The time between symptoms, arrival in hospital, and diagnosis of bleeding is crucial. The earlier the brain hemorrhage is confirmed and evaluated, the sooner a treatment plan can be made.

This paper presents a model based on deep learning techniques that can be useful for classifying ICH in patients from computer-tomography (CT) image slices. Deep learning [7] is a series of models, technologies, and architectures, based on Artificial Neural Networks, which has undergone a revolution in AI in the last years. Its doubtless success in real world problems reaches areas such as face recognition [8], music composition [9] or multilingual translation [10]. We refer the reader to [7] for a useful introduction to the basic concepts of deep learning. One of the most interesting areas of deep learning research is digital image processing, especially medical image processing. Among many other applications, we can cite disease classification [11], ROI segmentation [12] or medical object detection [13]. In this paper, we apply deep learning techniques to the study of ICH detection.

Among the disadvantages of using deep learning techniques in real-world problems we can cite the lack of a clear explanation. Although the accuracy achieved in many cases is high, experts find it difficult to extract knowledge from deep learning models and the results therefrom that justify decisions. To address this lack of explanation, various approaches have been published in recent years. One of these approaches is the field of visual explanations. Classification models of computational images incorporating such an approach usually provide an accurate classification, and they can also show the region of the image (the set of pixels) which is important in the classification process.

In this paper, we used the Gradient-weighted Class Activation Mapping (Grad-CAM) [14] method, with the aim of providing visual explanations via gradient-based localization. Beyond the use of Grad-CAM for visual explanations, this paper considers one of the most successful architectures for solving complex technical problems of neuronal networks (as vanishing/exploding gradients): the so-called residual neural networks (or ResNet, for short) [15]. The use of ResNet was integrated, in our model, with EfficientDet [16], which combines weighted bidirectional feature pyramid networks (BiFPN) with a compound scaling method (see below). From a technical point of view, to the best of our knowledge, this is the first time that ResNet, EfficientDet and Grad-CAM have been used together. The proposed model is not a simple juxtaposition of techniques, as described in the literature, but an integrated model.

On the one hand, it is worth mentioning that our approximation follows the current research guidelines on Green AI [17]. Roughly speaking, Green AI tries to reduce the carbon footprint of the current large computational processes by searching for a balance between consumed resources and the efficiency of the method. As is detailed below, our methodology obtained competitive results by using a training dataset and a deep learning network smaller than those of some competitors found in the literature. On the other hand, our methodology also includes visual explanations to understand model decision-making.

The aim of this study was to present an integrated deep learning model for the detection of intracranial hemorrhage in brain CT scans, together with a visual explanation system of decisions. The concept of "time is brain", used for ischemic stroke, could also be applied in this devastating disease, so automatic diagnosis, which takes seconds right after CT scan image reconstruction, can prioritize the images for evaluation by a radiologist or help in complicated or debatable cases.

## 2. Materials and Methods

### 2.1. Deep Learning

The first basic architecture used in our model was ResNet. The second was EfficientDet [16]. The main motivation for this architecture was to optimize the resources of the deep learning model for the detection of digital image objects.Usually, the great success of deep learning models is obtained only by expensive computers that can be used by large companies. Currently, many efforts are being made to adapt these models to simpler (and cheaper) hardware without losing accuracy. EfficientDet is a clear example of this line of research. The main contributions of using EfficientDet were two-fold. On the one hand, it uses a novel computing unit as the base of the model, the so-called weighted bidirectional feature pyramid network (BiFPN). On the other hand, the we proposed a compound scaling method for scaling up several features of the model (resolution, width, and depth), which led to a new family of object detectors, called the EfficientDet family, from D0 (the smallest one) to D7 (the largest one). According to the Green AI guidelines, we optimised our system in order to have competitive results with a minimum amount of resources, and, hence, only the D0 model, the smallest one, was considered. Resnet is inspired by pyramidal cells in the cerebral cortex, and tries to alleviate problems by introducing the so-called residual blocks. These blocks introduce a direct connection, which skips some layers in between. This connection is called the *skip connection* and is the core of residual blocks.

In our study, we propose a neural network architecture that combines EfficientDet [16] and ResNet [15] architectures. Technical details of ResNet and EfficientDet architectures can be found in Section A.1.

### 2.2. Grad-CAM

The Grad-CAM approach takes an image as input and uses the gradients of any output neuron in a classification network flowing into the final convolutional layer to produce a heatmap that highlights the relevant regions in the image to make the corresponding prediction. The authors argue that the last convolutional layers have the best compromise between high-level semantics and detailed spatial information. This method was also used in [18]. Grad-CAM is a form of post-hoc attention, i.e., it is a method applied to an already trained neural network. The basic idea behind this method is to take advantage of the spatial information that preserved through the neural network to understand which parts of an input image were important for the classification decision. Our model provides two different kinds of information:Firstly, our model provides an output as to whether the patient has ICH or not. Technically, the output is the probability of having ICH. The model outputs an affirmative answer if such a probability is greater than a fixed thresholdOur model further provides a color map on the input image where the red area corresponds to the pixels in the image which have been determinant in the decision.

We applied GRAD-CAM and obtained a heat map with focus on the ICH area. For the implementation of Grad–CAM, we used the *pytorch-grad-cam* library [19]. In addition, in our Grad-CAM experiments, we used enhancement smoothing, which, according to the library documentation, applies a combination of horizontal flips and image multiplications by [0.9, 1.0, 1.1], having the effect of better centering the CAM around the objects.

Figure 1 and Figure 2a illustrate the use of our model. An image where the model predicts bleeding (ICH) with a probability of 0.9897 is shown, and, in addition, Grad-CAM was able to determine the bleeding area to make that decision. Section A.2 provides a more technical description of this technique.

### 2.3. Dataset

Kaggle (https://www.kaggle.com/, accessed on 16 November 2022) is an online community of data scientists and machine learning practitioners that allows users to find and publish datasets, explore and build models in a web-based data science environment, work with other data scientists and machine learning engineers, and participate in competitions to solve data science challenges. The RSNA Intracranial Hemorrhage Competition [20] was a competition hosted by Kaggle at the end of 2019. This competition provides a high amount of annotated data, indicating if there is hemorrhage in the slice, including the corresponding subtype (subarachnoid, subdural, epidural, intraparenchymal and intraventricular bleeding).

The dataset contains 752,799 scan slices in Digital Imaging and Communications in Medicine (DICOM) format, from 18,938 patients. To avoid bias, since there were various slices from the same patient, the set of patients was divided into three randomly divided patients’ groups: train (90%, 17,044 patients), validation (5%, 947 patients) and testing (5%, 947 patients). It is important to note that there was a remarkable class imbalance, since the groups contained the following number of slices:Train: 97,525 with ICH, 580,934 without ICH.Validation: 5401 with ICH, 31,174 without ICH.Test: 5007 with ICH, 32,758 without ICH.

The ground truth for the Kaggle/RSNA data was taken from competition data. To mitigate this class imbalance, we randomly selected the same number of non-ICH slices as the ICH-sliced slices. This turned into 195,050 slices from 17,044 patients for training, 10,802 slices from 947 patients for validation, and 10,014 slices from 947 patients for testing. When evaluating our model on a patient level, we annotated the patients of the test set as patients with ICH in cases where any of the patient’s slices had ICH.

The CT scan slices provided us with 2D matrices corresponding to the scan in the Hounsfield unit (HU) scale. The HU scale is a linear transformation of the original linear attenuation coefficient measurement into one in which the radiodensity of distilled water at standard pressure and temperature (STP) is defined as zero HU, while the radiodensity of air at STP is defined as −1000 HU. In a voxel with average linear attenuation coefficient μ, the corresponding HU value is, therefore, given by:(1)$$HU=1000\times \frac{\mu -\mu_{\mathrm{water}}}{\mu_{\mathrm{water}}-\mu_{\mathrm{air}}}$$
where $\mu_{\mathrm{water}}$ and $\mu_{\mathrm{air}}$ are, respectively, the measured linear attenuation coefficients of water and air [21]. Thus, a change of one HU represents a change of 0.1% of the attenuation coefficient of water, since the attenuation coefficient of air is nearly zero.

As our neural network took three channels (i.e., a RGB image) as input, the scans with the HU matrices had to be pre-processed to obtain three matrices of size 512×512 pixels, one for each channel. To obtain these three matrices, three different windows were applied to every slice. Applying a window with *X* as lower bound and *Y* as upper bound meant that all HU values lower than *X* were converted to *X*, all HU values greater than *Y* were converted to *Y*, and the rest remained the same (see Figure 1).

We added a test group of 55 patients of clinical cases, 47 of them with intracranial hemorrhage and 8 of them as control, provided by Hospital Univeritario Virgen del Rocío (HUVR) and Hospital Universitario Virgen Macarena (HUVM), where two of the authors work. The controls were not healthy volunteers, but patients scanned by CT at admission for any other type of hemorrhage were excluded. The 55 patients were the maximum available patient number at the time. For our clinical data, the ground truth was taken from the available clinical data.

The images were evaluated for diagnosis per patient by experienced radiologists, neurologists and neurocritical doctors. They were not evaluated on a slice-by-slice basis, which is not done in clinical practice. The mean age of patients was 54 years, range [21...84]. There were 25 men and 30 women (55%).

### 2.4. Metrics

The metrics used to evaluate the experiments were accuracy, sensitivity, specificity, positive predictive value and negative predictive value, considering that the model predicted yes (slice with ICH) for a certain image slice if the probability was greater than 0.8 to reduce false positive predictions. Most binary models assume a cut-off value of 0.5, lower than ours.

### 2.5. Ethics

The research project in HUVR was overseen and approved by the Ethics Committee of the Hospital Universitario Virgen del Rocío in Seville, Spain (Cod. CEI2012PI/228). Informed consent was obtained from all patients participating in the study or from their close relatives. The images of the patients at the Hospital Virgen Macarena were obtained after they signed the informed consent form for CT, in which they accepted the potential use of the images for the purposes of promoting knowledge and research in relation to the disease from which they suffered.

## 3. Results

### 3.1. Kaggle Test Dataset

Table 1 shows the results of the evaluation of our model, called EffClass, with RSNA Kaggle competition test data and the application of different thresholds for the prediction of our model (i.e., the model predicted ICH if the probability returned by the model was higher than the threshold). The chosen thresholds of probability were 0.5, 0.7, and 0.9. The highest values are marked in bold.

The Kaggle dataset is a set of CT scan slices. However, the DICOM files of slices retain the information of the patient, and, therefore, we could have used these data to train our models at patient level, but we had to develop our own new criteria for evaluation if the *patient* was to be classified as having hemorrhage or not. This was outside of the available data of the Kaggle competition and we claim that if the patient´s CT scan had, for example, only one slice classified as having hemorrhage, that was not enough to classify and, most importantly, not enough to diagnose him or her positively as a whole patient having brain hemorrhage. Therefore, we evaluated the patients considering the following additional criteria: *The patient was diagnosed with hemorrhage if the model predicted, at least, n slices of the image with ICH being n a percentage (5% or 10%) of the number of slices, since the number of slices per patient varied*. We did not check for the continuity of the slices position, just the percentage.

For example, given two patients with 30 and 40 as the number of slices, respectively, and a threshold of 10%, the first patient would be classified as an ICH patient with ICH if the model predicted at least three (3) ICH slices, but the second patient would be classified as an ICH patient with ICH if the model predicted at least four (4) slices with ICH. The results of the experiments are shown in Table 2.

### 3.2. Test Results Obtained with Clinical Data

Despite the results that we obtained in the previous section being very promising, we needed to test our models with external data to validate that our models could be used not only with Kaggle data, but also with external data.

For this purpose, we evaluated the images of 55 patients from the Hospital Universitario Virgen Macarena and Hospital Universitario Virgen del Rocío patients. Both hospitals are in Seville, Spain.

In Figure 2a a slice with ICH is shown. It was quite difficult to see the ICH, but the Efficient Classification neural network predicted ICH with a probability of 0.9897. Figure 2b,c show the next consecutive two slices where ICH was clearly seen and the neural network predicted ICH with a probability of 0.9972 and 0.9997, respectively. Figure 2d shows a slice with ICH, where the hemorrhage was not detected, according to our threshold (0.8), and the probability was 0.7805. Figure 2e shows a positive slice where the probability was just above the threshold, 0.8101. On the other hand, in Figure 2f a slice without ICH is shown.

The results of the classification of the additional test 55 patients are provided in the last three rows or Table 2. All patients with ICH (47) were identified correctly. All control patients were identified as negative (8). There were no false positives diagnosed in control patients. NPV and PPV were 100% because there were no false negatives, nor false positives, at 10% of slices.

## 4. Discussion

Recently, some studies were published on the application of AI techniques to Intracranial Hemorrhage Detection, and many researchers have started to pay attention to the topic. Among many other studies, we can cite [23], where the authors used a fully convolutional neural network for classification and segmentation, with examination of ICH with computed tomographies. In [24] the InceptionV3 and DenseNet Deep Learning models were used for dealing with CT. In [25] the authors combined convolutional neural networks with other machine learning techniques to deal with ICH detection. In [26] the authors used a Dense U-net architecture for the detection of ICH. In [27] a novel deep learning technique, based on the Monte Carlo algorithm [28], was applied.

The paper by Voter et al. [22], published in April, 2021, deserves special attention. It is very important to note that we cannot make a fair comparison, since the datasets used in our experiments and the Aidoc experiments [22] are different. In the reference [22] dataset, only 9.7%, 349 of 3065 patients, were positive (i.e., patients with ICH), while in our test dataset, 39.2%, 371 of 947 patients, were positive. Furthermore, the results are highly influenced by the types of ICH that comprise the dataset and their proportion.

The study presented in [22] showed high accuracy (ACC) and negative predictive value (NPV). The highest accuracy value obtained from our experiments was 95.5%, with the Efficient Class model with threshold = 5% (overlapping confidence intervals). The study [22] showed the same true negative rate (TNR) as Efficient Class with threshold = 5%. With respect to the true positive rate (TPR), the Efficient Class model with threshold = 5% had a higher value (0.949). Finally, with respect to the predictive positive value (PPV), the highest value was 0.951, and it was obtained with the Efficient Class model with threshold = 10%.

A standard study design is needed to enable a rigorous and reproducible site-to-site comparison of new deep learning technologies. Our goal was to classify patients by means of their CT images as to whether they had ICH or not. This meant that we estimated a binary probability distribution, in which each image obtained the probability *P* that the patient had ICH and the probability 1−P that the patient did not have ICH. To this end, the original Kaggle dataset was pre-processed to train our model. In addition to many other technical details, pre-processing changes the classification label of all input data from the original label to a binary label, marking if the image shows if the patient has ICH or not. Bearing in mind that the target of the models presented to the Kaggle competition was to associate to each CT image a label with the type of ICH, not to classify if the patient had ICH or not, a comparison of the accuracy of our model and those presented in the competition was not possible.

A typical detected minimum size of a hemorrhage (width) by Aidoc software is shown in Figure 2a of [22] and was about 2 mm. In clinical practice, there are rarely less than 25 slices per brain scan. We did not check or train the model for the size of the hemorrhage. Let us note that the model was trained on slices with yes/no for all the slices, and we did not train the model for the location of the hemorrhage. The Grad-CAM map should not be mistaken for the location of the hemorrhage. It can be interpreted as the usual location of the hemorrhages and the pixels considered in the evaluation of the presence of hemorrhages for this slice, but not as a precise location.

In paper [22], the authors had the objective of determining the diagnostic precision of an AI decision support system (DSS), developed by Aidoc [29], in diagnosing intracranial hemorrhage (ICH) in non-contrast head CT and to assess the potential generalizability of an AI DSS. This retrospective study included 3605 consecutive, emergent, non-contrast adult head CT scans.

Each scan was assessed for the ICH by a certified neuroradiologist and Aidoc. The authors determined the diagnostic accuracy of the AI model and performed a failure mode analysis with quantitative CT radiomic image characterization. Of the 3605 scans, 349 cases of ICH (9.7% of studies) were identified. The neuroradiologist and Aidoc interpretations were consistent in 96.9% of the cases and overall sensitivity, specificity, positive predictive value, and negative predictive value were 92.3%, 97.7%, 81.3% and 99.2%, respectively.

Aidoc was wrong in 101 of 3605 studies, which corresponded to an accuracy of 97.2%. This study revealed decreased diagnostic accuracy of an AI DSS at their institution. Despite its extensive evaluation, they were unable to identify the source of this discrepancy, raising concerns about the generalizability of these tools. Our method had some higher metrics (ACC, TPR and PPV) than Aidoc FDA submission, bearing in mind that the amplitude of the confidence intervals of Aidoc FDA submission was wide.

In [30], the authors also used the Kaggle dataset and compared their results with clinical data available to them, keeping the classification at slice level without applying their model at patient level. These results further highlighted the need for a standardized study design to allow a rigorous and reproducible site-to-site comparison of emerging deep learning technologies.

The irruption of deep learning as a new set of AI techniques represents a major challenge in medical applications. This type of technology, after CE (Conformité Européenne, in French) security marking, Food and Drug Administration (FDA) or respective national agency clearance, can be very useful for centers, especially small community hospitals, where there are no radiologists on call to interpret these images and that depend on non-specialized physicians.

In many fields, the expertise of radiologists is key to obtaining a correct diagnosis. In this case, the use of the latest technology for computer image classification in medical images is a major contribution to human expertise.

In fact, specific diagnostic support software is already used in ischemic stroke in a wide range of hospitals. In this context, this paper presented EffClass, a deep learning model that combined two of the most successful models for image classification, and we showed that our model was competitive with some of the current most widely used commercial, CE certified and FDA approved methods, such as Aidoc [29] or e-ASPECTS [31].

Diagnosis of ICH has been the subject of several papers and is a field of intensive research using traditional techniques [4] and deep learning [23,24,25,26]. The solution [32] has AUC 0.846 (CI 0.837–0.856). Another is BRAINSCAN.AI [33] which has AUC/ROC of 0.976 for hemorrhages. It is a CE-certified solution and is now installed in more than 40 hospitals in Poland [34].

However, the use of neural network methods in medicine, as a support for decision processes, has an important drawback. Neural networks are usually considered as black boxes, where high accuracy is not considered, due to the lack of explanations. Human experts want to know not only the decision of the AI system but also the motivation of such a decision. In such a way, our model not only provides high accuracy on the decision, but it also provides visual explanation with an intuitive color map on the original image. The Grad-CAM method was used to highlight the area of the input image that was relevant in the decision. This mixture of high accuracy together with visual explanation ensures that the proposed model is one of the most competitive among current models.

The limitation, as of today, is that the method is for research use only and the command line interface with PyTorch is comfortable only to IT specialists. It requires the extraction of the DICOM hospital image from the hospital system to a desktop computer.

Future research lines can be considered. From the application point of view, the deep learning technologies proposed in this paper could also be applied to obtain classification and visual explanation of other medical decisions based on medical images. From a technical point of view, deep learning is continuously evolving, and other recent technologies on classification and visual explanation deserve to be explored, including 3D approaches.

Despite the widespread use of deep learning in medical image classification, this paper makes a two-fold contribution to this field. On the one hand, the accuracy achieved is competitive with the AI techniques most commonly used in ICH classification. This represents an advance over many papers found in the literature. On the other hand, many deep learning tools are black boxes providing classification without explanation. In this paper, we proposed the use of visual explanation technology to provide information on the key areas of CT. This additional information complements other deep learning tools and makes our proposal a useful tool for clinicians.

Our code is available online (https://github.com/Keredu/Intracranial-Hemorrhage-Detection, accessed on 16 November 2022). More details of the framework can be found in Section A.3. Let us recall that the CAM techniques for obtaining visual explanations of Deep Learning decision were previously used by other researchers (see, e.g., [35] or [36]), but, to the best of our knowledge, this is the first time that ResNet, EfficientDet and Grad-CAM have been used together. The combination of such technologies is a contribution of this paper that can shed light on future research in this area.

## 5. Conclusions

If not treated immediately, ICH can cause the death of a patient. The time between symptoms and diagnosis is crucial. In this way, current AI methods can help to shorten this time.

In this paper, we presented a deep learning model that can detect bleeding in CT scans of a patient in DICOM 2D image files. This model is highly competitive.

Furthermore, the model not only provides classification of CT scans, but also gives information about the region of the image which motivated the decision of the AI model. This should not be confused with the location of the diagnosed hemorrhage. In this way, our proposed model could be used as an assistant for the diagnosis based on CT scan and, in addition, it could provide information that could be useful to human experts in making their own decisions.

The results obtained could be a starting point for further research for the integration of current technologies of explanations of Artificial Intelligence methods (the so-called Explainable Artificial Intelligence (XAI)) within medical research.

## Figures and Tables

**Figure 1 jimaging-09-00037-f001:**
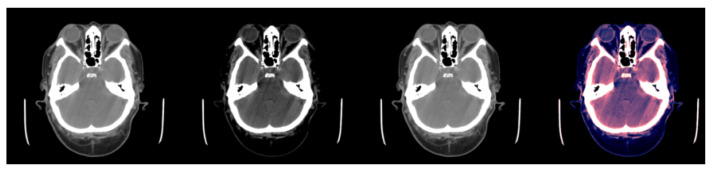
Example of slice windowing as pre-processing of an HU matrix to obtain three different matrices (window used between brackets). From left to right: (1) slice with [0.80], (2) slice with [−20, 180], (3) slice with [−150, 230], (4) the three windowed slices stacked (each channel is associated to one color RGB for visualization purposes).

**Figure 2 jimaging-09-00037-f002:**
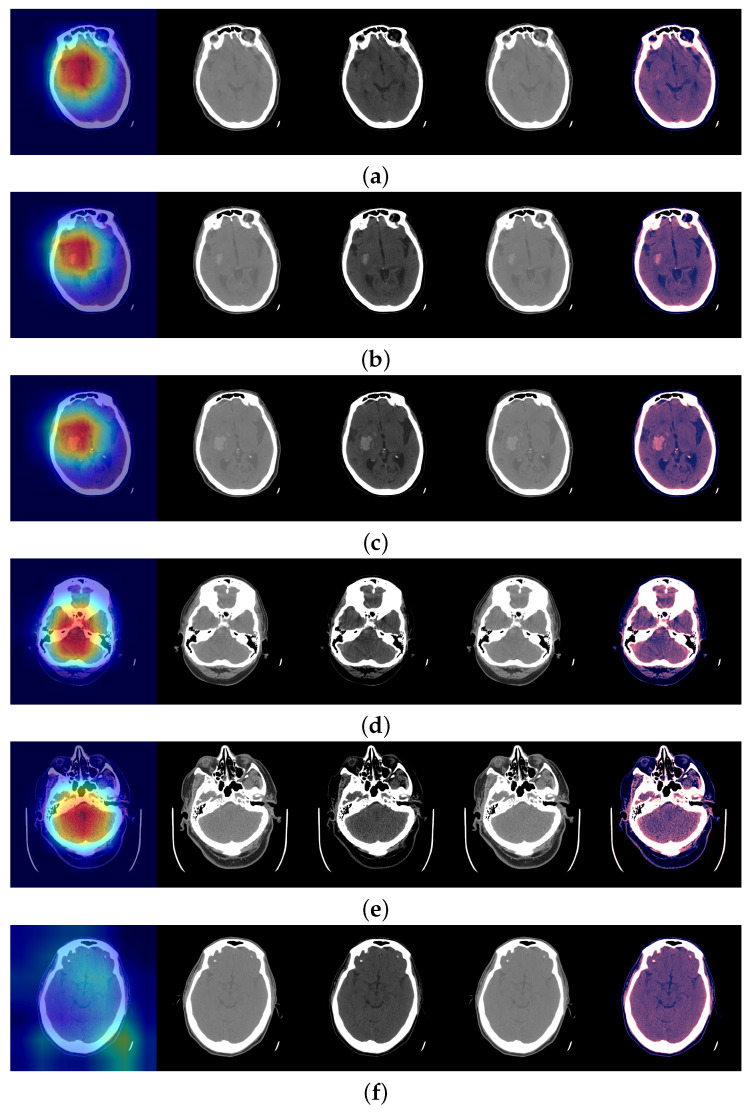
The model Efficient Classification predictions were all correct. In addition, Grad-CAM was able to focus on the bleeding area to make that decision. In each image, from left to right: Grad-CAM, [0.80] window, [−20, 180] window, [−150, 230] window, the three windows stacked. From (**a**–**f**) the probabilities of ICH in the slice are 0.9897, 0.9972, 0.9997, 0.7805, 0.8101 and 0.0097.

**Table 1 jimaging-09-00037-t001:** Results per model and threshold (**TH**). Higher metrics values remarked on. **EffClass**: Efficient Classification, **ACC:** Accuracy, **TPR:** Sensitivity (Recall), **TNR:** Specificity, **PPV:** Positive Predictive Value (Precision), **NPV:** Negative Predictive Value, **PR:** Precision–Recall AUC, **ROC:** ROC-AUC.

Model (TH)	ACC	TPR	TNR	PPV	NPV	PR	ROC
EffClass (0.5)	**0.927**	**0.914**	0.94	0.938	**0.916**	**0.979**	**0.978**
EffClass (0.7)	0.918	0.872	0.965	0.961	0.883	**0.979**	**0.978**
EffClass (0.9)	0.881	0.774	**0.987**	**0.984**	0.814	**0.979**	**0.978**

**Table 2 jimaging-09-00037-t002:** Results per model, with at least *n*% of the slices with ICH. The best metric per model and threshold (**TH**) are marked in bold. **EffClass**: EfficientClassification, **ACC:** Accuracy, **TPR:** Sensitivity (Recall), **TNR:** Specificity, **PPV:** Positive Predictive Value (Precision), **NPV:** Negative Predictive Value, **N:** Number of patients, **ICH (%):** Percentage of patients with ICH. (*): No confidence interval provided. (**) Confidence interval impossible to calculate.

Model (TH)	ACC	TPR	TNR	PPV	NPV	N	ICH (%)
Aidoc (FDA/501 k) [22]	92.9% (*)	93.6% (86.6–97.6)	92.3% (85.4–96.6)	91.7% (84.9–95.6)	94.1% (88.4–97.2)	198	47.5%
Aidoc [22]	97.2% (*)	92.3% (88.9–94.8)	97.7% (97.2–98.2)	81.3% (77.6–84.5)	99.2% (98.8–99.4)	3605	9.7%
EffClass (5%)	95.5% (93.9–96.7)	94.9% (92.1–96.9)	95.8% (93.9–97.3)	93.6% (90.8–95.6)	96.7% (94.9–97.8)	947	39.2%
EffClass (10%)	93.6% (91.8-95.0)	88.1% (84.4–91.3)	97.0% (95.3–98.3)	95.1% (92.3–96.9)	92.7% (90.6–94.4)	947	39.2%
EffClass test (2.5%) 55 patients	94.6% (84.9–98.9)	100.0% (92.5–100.0)	62.5% (24.5–91.5)	94.0% (86.5–97.5)	100.0% (**)	55	85.5%
EffClass test (5%) 55 patients	96.4% (87.5–99.6)	100% (92.5–100)	75.0% (34.9–96.8)	95.9% (87.6–98.7)	100.0% (**)	55	85.5%
EffClass test (10%) 55 patients	100.0% (93.5–100.0)	100.0% (92.5–100.0)	100.0% (63.1–100.0)	100.0% (**)	100.0% (**)	55	85.5%

## Data Availability

Data from Kaggle RSNA Intracranial Hemorrhage Competition are publicly available and can be obtained from [20].

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
