# Peer review of "Deep Learning Applied to Intracranial Hemorrhage Detection"

_2313-433X, 2023, doi:10.3390/jimaging9020037_

Round 1
Reviewer 1 Report (New Reviewer)
The submitted paper handles an important topic and introduces a promising approach of intracranial hemorrhage detection with low time latency and shows good processing results. The paper us well written but especially article structure and medical background need revision.
Following issues, I like to note:
Consider a classic subsection structure. The subsection “Deep Learning in the Detection of Intracranial Hemorrhage” is a mixture of introducing sentences, material and methods information, explanation of the term “Deep Learning” and preliminary works. I recommend splitting this section and restructure the article in just the classic 5 subsections.
In this context suggest restructuring “Materials and Methods” section f.e. in (a) Deep Learning including the main points of lines 124-145; (aa) ResNet; (ab) EfficientDet; (ac) BiFPN; (b) Grad-CAM (c) Dataset (d) Metrics / Statistics …
Your explanation of division of intracranial hemorrhages is false. Subarachnoid bleeding (SAH) is one subtype of ICH. It is not a dichotomous differentiation between ICH and SAH. SAH, subdural bleeding, epidural bleeding, intraparenchymal bleeding and intraventricular bleeding are the 5 subtypes of bleedings in cCT summed to intracranial bleeding/intracranial hemorrhage.
As I understand your AI tool does not separate this subtypes so it is not necessary to subdivide the ICH.
I) Introduction
This subsection should be condensed. It shows redundancies like the introduction of Grad-Cam (Lines 52-63) and explanation of the used neuronal network (Lines 64ff). This information should be given in the introduction but significantly shorter, explanations are given in “M&M”-section.
There is a hard jump from “what is an ICH” (Lines 1-30) to AI (Lines 31-77) to CT (78-86) and back to clinic (Lines 87ff). CT part could be switched to “M&M”. If starting with clinical background continue with information given in Lines 87ff. Please not that there are redundancies; giving surgery methods (Lines 95-98) and therapeutical treatment (Lines 101-193) is not necessary. Main points are Lines 98-100 “Time is Brain” and Lines 104-106 “early diagnostic = good outcome”.
Note: Not just spontaneous ICH (according to stroke) are relevant you should also consider traumatic bleedings.
The sentence […] and they can also show the region of the image … (Lines 49-51) is very important because this is a big problem in handling AI results in clinical usage. This visualization is one of the main points/improvements of this work!
Lines 107-112 are not necessary.
There is no written aim of study.
II) Deep Learning for the Detection of Intracranial Hemorrhage
Like noted above consider dissolving of this section.
Lines 114-123 and 146-149 are introducing sentences, these could be modified and used in “Introduction” section.
The rest of this section has textbook character and could be considerably shortened. You may use citations of articles which give further explanation.
III) Materials and Methods
Consider subsections like mentioned above to improve clarity.
3.1 Deep Learning:
Restructure this subsection using the moved content (Sections 1. Introduction and 3. Deep Learning […]) and add short explanation of ResNet and EfficentDet.
3.2 Dataset:
Consider switching of paragraphs 1 and 2. Lines 178/179 are not necessary.
512x512 matrix should be usually used in cCT scans, why pre-processing?
The slice thickness is an important influence factor. For my own experience the AI results differ with different slice thicknesses. Please give the count of slices with thickness of 4/3/3/1 mm of the Kaggle data set and of your own dataset.
Classification in Line 207 is incorrect. A subarachnoid bleeding is one of the 5 subtypes of an intracranial bleeding. By this reason a separation of intracranial bleeding and subarachnoid bleeding is not senseful.
Abbreviations “HUVR” and “HUVM” are not explained.
Ethic statement (Lines 210-216) should be given in separate subsection (3.XY).
Who has generated the ground truth for the evaluation per slice and how?
3.3 Grad-CAM:
Lines 233/34 – What is shown if there is no bleeding detected. Is there any output of the Grad-CAM or just standard CT image without red marker?
Last paragraph (Lines 246-250) should be moved to “Discussion” section.
3.4 Metrics:
Are the accuracy parameters calculated one slice level only or also on patient level? For usage in clinical conditions especially the results per patient are relevant.
IV) Results
Lines 264-270 – This paragraph needs more background in “M&M section. How was the ground truth generated for a slice-by-slice evaluation. Who has read the cCT images with what experience?
In my opinion the additional criteria given in Lines 271-273 is not useful. The “n slices of image stack” modus depends on slice thickness. I know that a threshold is needed to classify as bleeding pos. or neg. but 10% is far to high for clinical use. A cCT contains around 120 slices with 1 mm slice thickness and 40 slices with 4 mm slice thickness. This results in 12 or rather 4 blood containing slices and a distance of > 1 cm. But an ICH of smaller size is also relevant. The 5% cut off may be o.k. but 2.5% to 3% would be desirable (bleeding size of at least 3-4 mm).
Also, the continuity is a relevant information to calculate the likelihood of a bleeding. Continuous positive images are more likely a true/relevant bleeding than slices with gap (except diffuse subarachnoid hemorrhage).
You should calculate at least accuracy of your KI using 2.5% or 3% as cut off. Also consider implementation of a rating of the continuity in further improvement of the AI tool.
As far as I can remember, AiDoc evaluates their results on patient not on slice level and even with smaller bleeding sizes than your cut offs. By these reasons the results given in table 2 are not quite comparable.
The additional sentence to Fig 3 (Lines 287/288) may be correct but the used window settings are not optimized for bleeding detection (f.e. C 60 / W 100; HU blood ~ 60HU). Also show this setting or just remove the sentence. The highlighting by Grad-CAM is an important point but not for bleeding detection by the human reader but to understand the decision-making process of the AI.
Results given in Lines 296-298 may be impressing but the cut-off biased the significance. Please, also calculate 5% and 2.5/3% cut-off.
What is the processing time of your AI tool?
V) Discussion
Paragraph 2 Lines 309ff: The results of the citied published studies are not directly comparable because of different methods f.e. slice or patient evaluation; different/one slice thickness; threshold for positivity. On other important point are different results for development conditions and real-world data (f.e. AiDoc). Your results using Kaggle are comparable maybe slightly better but the results giving for real-world data are 1) not comparable because of different patient population and 2) yours seem to be overrated maybe because of the high cut-off of 10%.
Available FDA-certified AI tools for ICH detection show similar result.
Is your AI tool able to classify the ICH subtype? If not, is this ability planned in future?
Lines 373-377 deal with a different subject – ischemic stroke. There are also available supportive tolls for bleeding detection to quote.
Time latency of manual reading have to be discussed and depends on the conditions (Lines 378/379). This statement is not necessary because you do not report processing time of your tool.
VI) Conclusion
Please formulate a clear conclusion.
Paragraph 1: This comparison to AiDoc is not possible. Your results/methods do not support this statement. Lines 410-413 are explanation of the method and should be moved/deleted.
Paragraph 2: This statement is false. Like given above - subarachnoid bleeding is one of the 5 subtypes of intracranial hemorrhage. It is not necessary to subdivide the ICH because your tool decides ICH yes/no and don’t give the subtype.
Lines 415-420 should be moved to “Introduction” of “M&M” subsection.
Lines 421-428 should be moved to “Discussion”.
Figures
Figures 1 - 3 need revision of the figure captures. Parts of the captures are given in text.
Author Response
We provide the answer to the reviewer in the attached PDF file. Our answers are typed in black or blue bold text with the "Answer: " preceding it directly in the text of your comments. We thank you for the comments and remarks that made the article of a higher quality.

Reviewer 2 Report (New Reviewer)
The article discusses about Deep Learning approach applied to Intracranial Hemorrhage Detection. The paper is well presented. The main question addressed by this paper is Intracranial Hemorrhage classification using deep learning techniques. It is relevant and interesting. The method is based on EfficientDet. The authors also provided visualization using GradCAM.
I have the following concerns:
1. Is equation (1) s proposed by the authors? If not then cite the source of it.
2. In table 2, a comparison with other approaches is shown. The value of N is varying. The authors of present article have used N=55. It is recommended to compare with other approach where N=55 is considered for the experimentation.
Author Response
We provide the answer to the reviewer in the attached PDF file. Our answers are typed in black or blue bold text with the "Answer: " preceding it directly in the text of your comments. We thank you for the comments and remarks that made the article of a higher quality.

Round 2
Reviewer 1 Report (New Reviewer)
The quality of your paper was significantly approved.
In my opinion the introduction section is still too long / overloaded, but its suitable for publishing.
This manuscript is a resubmission of an earlier submission. The following is a list of the peer review reports and author responses from that submission.
Round 1
Reviewer 1 Report
The authors present their work using deep learning algorithms to classify intercranial hemorrhage (ICH) using both a publicly available large dataset and clinical data acquired from patients outside of this large database. The results clearly show that their implementation for the classification of ICH is working with high accuracy (92.7%).
I cannot judge its novelty, but the addition of representation of the pixels that contributed significantly to the classification is appreciated. The format in which the work is presented would benefit from a major revision, both regarding the structure, the style and language used and the scientific content. I cannot recommend publishing this work without the major revision.
Comments on structure of the manuscript
This manuscript would largely benefit of some restructuring and rewriting. Section 1, 2 and 3 are written in a review like format. I appreciate the level of information provided but feel that for a scientific report (article), the first three sections should be shortened and combined into one introductory section.
A large part of section 4, should be placed within the introduction as well. Even this section is more appropriate for a review report, but not as such for an article format.
Part of section 5. Results should be moved to the Material and methods section. For example the metrics used to evaluate the experiments should not be mentioned in the results section. Even lines 273-280 should be moved to material and methods. Remove header section 5.1 (L302) or rename it to 5.2 and name the first part of the results 5.1
The conclusion section is lacking (as required by the journal instructions), and I would like to see a more clearly formulated conclusion of the work.
Comments on the scientific content
My interpretation is that the authors aim to accurate classify CT images with and without ICH. I would expect that their implementation would be benchmarked against the contributions in the RSNA, Intracranial Hemorrhage Detection challenge with almost 1300 contributions using the same dataset as the authors have used (https://www.kaggle.com/c/rsna-intracranial-hemorrhage-detection/leaderboard). At the homepage for this challenge, the top contesters are listed with the performance of each implementation. It is crucial to discuss the authors results on the same dataset with what has been achieved in the challenge. And if not, make clear why such a comparison is not appropriate.
The goal of the AI tool is to identify the patients that have ICH. After reading section L273-280, I realize that it is not trivial to decide how many slices should be classified positive for ICH to diagnose the patient with ICH. I have the following suggestions/comments:
- Does the slice thickness vary between different datasets? Is it certain to assume that a patient that is scanned with 40 slices has a 75% less slice thickness than the patients scanned with 30 slices? This is relevant to know if one decides to demand more positive slices from a scan with 40 slices than with 30 slices.
- Is it a requirement that in case 4 slices need to be positive that these slices are continuous, e.g. slice 26,27, 28, and 29. Or does this mean that any 4 slices can be positive for indication of ICH?
- What is the minimal size of ICH that a radiologist can identify? Should this define the number of slices that should be positively identified with ICH.
- Do the results contain any positive slices of ICH, in which ICH was wrongly classified because the minimum number of slices was not met?
Specific points
Line 22: ‘once brain cells die, they do regenerate’. Any cell that dies will not regenerate, but tissue has regenerative capacity. Even brain has this capacity. Please reformulate and add references.
Line 28: ‘Classifying ICH in both patients and CT image slices’. -> the authors analyze images in both cases for presences of ICH. One set is part of the large public database, the others from two collaborating hospitals. Please reformulate.
Line 316: ‘it is difficult to see’ -> please pick an example that can be identified by a trained radiologist, but not by most people. If the lesion can be seen in this figure by me in a PDF file using a normal computer screen, the lesion would certainly not be missed by a trained radiologist using dedicated viewing stations.
Author Response
Please see the attached file. We have put the replies to editor´s text in bold inside his/her text.

Reviewer 2 Report
The reviewed paper "Deep Learning applied to Intracranial Hemorrhage Detection” addresses an important area of AI-based diagnosis of Intracranial Hemorrhage (ICH) from CT images of the brain. The authors propose to use a few well-known Deep Learning architectures such as ResNet and EfficientDet to classify individual CT slices from Kaggle competition as containing ICH or not. In order to make their approach interpretable, the authors added a well-known gradient class-activation map (Grad-CAM) technique. Grad-CAM produces heat maps that roughly correspond to parts of input images responsible for the model’s predictions. The obtained results were evaluated using classical machine learning metrics and compared to the results of similar research published earlier. The authors could be complimented on making an effort in explaining all the relevant background from both IT as well as medical specialist perspectives. Comments: The novelty of this research is questionable and needs to be clarified. Neither of the components described in the manuscript (ResNet, EfficientDet, Grad-CAM) is new nor is the idea of applying the deep neural network to medical images to diagnose a disease X is new and then use class activation method to visualise it (e.g. https://link.springer.com/chapter/10.1007/978-3-030-59137-3_23). Hence it for future submissions, it is highly recommended to highlight this point very early on in the manuscript, i.e. in the abstract and introduction. To this end, a deeper and more comprehensive literature overview might help to grasp better the current research landscape. Claims are not well-justified. At present, the authors took a large dataset from the public domain (i.e. available on Kaggle) and compared their results to one competitor algorithm (from AiDoc), which is absolutely insufficient amount of evidence needed to justify the following claim made in the introduction by the authors: “On the one hand, we present a Deep Learning architecture that, in some sense, improves the SOTA methods for classifiying ICH". One may (and should) ask how the presented approach compares to the performance of the best models submitted in the same Kaggle competition as data originates from. More than 1000 teams have competed in the competition, and if the model developed by the authors would perform better according to the chosen metric, it is worth being considered a new SOTA in this task. Also, it is important to note here that having five samples in the validation dataset is not enough to make sound arguments about the algorithm's performance. It does not help that Grad-CAM's visualization is identical for three out of 5 cases. Presentation requires significant improvement. The manuscript contains an unacceptably high number of grammatical, syntactical, and semantical mistakes. Suboptimal or wrong word order in many sentences, missing propositions, spelling mistakes make text very hard to read and comprehend. Here are a few specific examples:
1. "Each scan was evaluated for ICH by both a certificate of added qualification certified neuroradiologist ..." - this is unclear what "a certificate of added qualification certified" mean. 2. "The severity and outcome of a ICH depends on ... brain location, ..." 3. "For example, some models provide a person’s name if a face is given ..." - should be reformulated as "some models are capable of generating a person's name from their face". There are a lot more problems in the text that must be corrected. The general recommendation is to do very thorough proofreading of the text before the next submission. Some figures could have been combined to make better use of space, e.g. Figures 1 and 2, if kept in the manuscript, might have been two panels in one figure. Also figures 4, 5, 6, and 7 should have been combined into one figure, at the moment their legends are very repetitive and information is almost the same. The structure of the manuscript as well as its content of some sections should be revisited. Something I have complimented the authors for, namely taking care of thoroughly explaining the background, was taken to the extreme in the Introduction and Materials and Methods sections of the manuscript. The readers will be swamped by the excess of information provided about Deep Learning, ideally, the paper should only focus on the most important bits that are absolutely necessary to understand the results and the discussion that is to come. At present, it seems that explaining the very basics of Deep Learning in such a paper is way beyond its scope and should be cut down. The same is relevant for paragraphs about ResNet, Grad-CAM and Kaggle. Please, move metrics from the Results section to Materials and Methods. Also, please, discuss the reasons these metrics have been chosen to evaluate the methods. The Materials and Methos section should contain a lot more information about the proposed model, not only about ResNet, EfficientDet and Grad-CAM.
Author Response

(The authors gave the same response as above.)

Reviewer 3 Report
In this manuscript, Cortes-Ferre et. al. introduced a deep convolutional neural network model that can predict intracranial hemorrhage based on CT scan of the brain. The dataset size was decent and the model’s performance was exceptional. The text was well-written with ample amount of background introduction. However, I have a major concern about the generalizability of the model and some minor concerns.
The model was trained, validated, and tested on a Kaggle challenge dataset with a decent amount of data. In terms of real clinical cases, however, they provided 5 clinical samples to prove the generalizability of the model. To make the argument stronger, the readers may expect a larger set with a larger variety of cases. It would be better to also show some “corner cases”, for example, the cases that physicians might also be confused with.
In addition, intracranial hemorrhage generally has a very obvious pattern on the CT images as shown in the figure4-6 (a white region). So, it is not surprising to see a good performance of the deep learning model to capture such a simple and obvious feature. This leads to the fundamental question of why do we still need AI to step into this specific task? I think the authors should have a better explanation.
Also, the authors mentioned that their model is a combination of Resnet and EfficientDet. As a technical reader, I would be willing to see a diagram/figure showing how they did the integration of these 2 classical architectures.
And perhaps, the grad-CAM visualization could also be applied to the 3 windows separately. Also, maybe talk about the relative contribution/importance of each window based on the models’ weight matrices.
Author Response

(The authors gave the same response as above.)

Round 2
Reviewer 1 Report
I thank the authors for their responses on my review report. I have new further comments or questions and would recommend publication in its current form.
Reviewer 2 Report
Dear authors,
thank you for your response and thank you for taking time to address some of the issues pointed out in the previous review.
Here are a few thoughts on your response to my comments:
You claim that the main contribution of your work is two fold: a) your model achieves performance comparable to other AI methods in the field b) use of Grad-CAM. The first claim seems to lack evidence as the only method you compare yourselves with is Aidoc, trained as you say it yourselves on a different data, hence is not comparable to your algorithm directly. In the review I asked you to compare your model to Kaggle competition top submissions, which to my best knowledge was not done and even mentioned in the response. The second claim is neither very helpful as researchers in medical imaging have been using Grad-CAM and other similar visualisation techniques (e.g. Score-CAM) long before your work. I gave a link to a publication in my review.
